

# Defining metrics of the Quasi-Biennial Oscillation in global climate models

Verena Schenzinger[1], Scott Osprey[2,1], Lesley Gray[1], and Neal Butchart[3]

[1]Atmospheric, Oceanic and Planetary Physics, University of Oxford, UK
[2]National Centre for Atmospheric Science, UK
[3]Met Office Hadley Centre, Exeter, Devon, UK

*Correspondence to:* Verena Schenzinger (schenzinger@atm.ox.ac.uk)

**Abstract.** As the dominant mode of variability in the tropical stratosphere, the Quasi-Biennial Oscillation (QBO) has been subject to extensive research. Though there is a well developed theory of this phenomenon being forced by wave-mean flow interaction, simulating the QBO adequately in global climate models (GCMs) still remains difficult. This paper presents a set of metrics to characterise the QBO using a number of different reanalysis datasets and the FU Berlin radiosonde observation dataset. The same metrics are then calculated from CMIP5 and CCMVal-2 intercomparison project simulations which included a representation of QBO like behaviour to evaluate which aspects of the QBO are well captured by the models and which ones remain a challenge for future model development.

## 1 Introduction

After being referred to as "mystery or freak" by one of its discoverers (Reed, 1967), the Quasi-Biennial Oscillation (QBO) now is accepted as the dominant pattern of variability in the equatorial stratosphere (e.g. Baldwin et al. (2001); Pascoe et al. (2005)). Between 3 and 100 hPa, zonal wind at the equator is characterised by a pattern of descending easterly and westerly shear zones, with wind direction changing about every 14 months (see, for example, the ERA-Interim and observations in Figure 1). The earliest regular observations of the equatorial stratosphere and hence the discovery of the QBO is credited to Ebdon (1960) and Reed et al. (1961). Angell and Korshover (1964), who named the phenomenon the "Quasi-Biennial Oscillation", pointed out oscillatory behaviour not only in zonal wind, but also in temperature, total ozone and tropopause height. The regularity of the oscillation makes it the most repeatable mode of variability in the atmosphere, beyond the diurnal and seasonal cycles. Whether or not the QBO remains as regular under future climate change is an outstanding question (Osprey et al., 2016; Newman et al., 2016).

Early attempts to explain the driving mechanisms of the QBO failed in describing one or more of its main features, such as the quasi-biennial periodicity, the downward propagation or the roughly constant amplitude during the descent. Initial thoughts regarding the driving processes involved internal feedbacks, natural atmospheric modes, an unknown external process or a combination of those (Baldwin et al., 2001). The first study to explore possible forcing by gravity waves was by Lindzen and Holton (1968). They showed that vertically propagating waves could provide momentum for the QBO. This theory of wave-mean flow interaction was supported by a laboratory experiment, carried out by Plumb and McEwan (1978). They were



able to produce a descending oscillation of the mean flow in a large annulus containing a salt-stratified fluid, the first practical demonstration of a laboratory analogue for the QBO.

With the development of a theory of equatorial waves in the late 1960s, that was observationally confirmed (Maruyama, 1967; Wallace and Kousky, 1968), the work of Lindzen and Holton (1968) could be refined. Holton and Lindzen (1972) simulated

a QBO-like oscillation in a simple one dimensional (1D) model, driven by vertically propagating Kelvin and Rossby-gravity waves that contribute westerly and easterly momentum forcing, respectively.

The first successful simulations of a realistic QBO were achieved in a 2D model by Gray and Pyle (1989) and in a 3D global climate model by Takahashi (1996). Follow on studies describing simulations that captured a QBO signal were Horinouchi and Yoden (1998); Takahashi (1999); Scaife et al. (2000) and Hamilton et al. (2001). Adequate simulation of the QBO de-

pends, amongst others, on resolution (horizontal and vertical), parameterised gravity wave forcing from sub-grid scale waves (Giorgetta et al., 2006) and placement of the model lid (Lawrence, 2001; Osprey et al., 2013). However, there is not a simple model configuration that would guarantee a successful QBO simulation and despite there being a well established theory of the QBO, not all climate models can produce it. Of the approximately 30 models submitted to the Coupled Model Intercomparison Project 5, CMIP5 (World Climate Research Programme, 2010), only four have a QBO signal (Lott et al., 2014). In the models

submitted to the Chemistry-Climate Model Validation Activity (SPARC CCMVal, 2010) there are five out of fourteen, with three of them variants of the Met Office Unified Model (Butchart et al., 2011).

The aims of this paper are to establish a set of standard metrics that comprehensively characterise the QBO. Using these characteristics, the performance of 10 historical model simulations is assessed and compared to observations and reanalysis datasets as the starting point of the World Climate Research Programme's (WCRP) Stratosphere-troposphere processes and their role

in Climate (SPARC) QBO initiative (QBOi[1]). An additional purpose is to provide a benchmark for the current status of the representation of the QBO in global models against which the future QBO simulations can be evaluated.

## 2   Data

For this study, monthly means of zonally averaged zonal wind and temperature of four CMIP5 and five CCMVal-2 models as well as one from CMIP3 that internally produce a QBO were investigated. Table 1 lists these models and further details.

Model data were obtained from the British Atmospheric Data Centre (BADC[2]). For comparison, the Berlin dataset (Freie Universität Berlin, 2015) of equatorial zonal wind from radiosonde observations covering 1956 to 2015 (Canton Island 1956-1967, Gan/Maledive Islands 1967-1975, Singapore 1967-2015) was analysed, as well as several reanalysis datasets (Table 2) made available through the SPARC Reanalysis Intercomparison Project (S-RIP) project[3]. When an average of more than one reanalysis was used, only the three relatively recent products (ERA Interim, Merra, JRA55), comprising the years 1979-2009,

were employed.

---

[1]http://users.ox.ac.uk/~astr0092/QBOi.html

[2]http://badc.nerc.ac.uk/home/index.html

[3]http://s-rip.ees.hokudai.ac.jp/





## 3 Definition of characteristic metrics

With an increasing number of climate models resolving the stratosphere and showing an oscillation in equatorial zonal mean zonal wind, the incentive of comparing the quality of these simulations arose. One obvious metric, the period of the oscillation, was quickly established. However, the period is not the only characteristic of the QBO; the oscillation has a structure in

latitude and height and the behaviour of easterly and westerly shear zone differs. Furthermore, it is not a typical oscillation with one constant restoring force, which leads to a variety of periods (Dunkerton, 2016). There might be an interaction with the semiannual oscillation or the 11 year solar cycle as well as the annual cycle in the troposphere that can influence timing of phase changes and descent of the shear zones. To assess these different aspects that are seen in the zonal wind observations (Figure 1), we propose a set of characteristic metrics, including, for example, the height of the maximum amplitude, the latitudinal and

vertical extent and descent rates of each shear zone (Table 3, 1st row).

Figure 2 shows the process of metric derivation using the reanalyses mean (ERA-Interim, MERRA, JRA55) as an example. Derived values from the individual reanalyses, the FU Berlin dataset and model simulations are provided in Table 3. The Fourier transformation of the equatorial zonal mean wind field (Figure 2, left panel) is used to determine the height of the maximum QBO amplitude. At this height, the timeseries of $\bar{u}$ is used to find the QBO period, defined as the time between

every other phase change (Figure 2, right panel). The months in which these phase changes occur are used to look for annual synchronisation of the QBO. The mean amplitudes of the easterly/westerly shear zones are extracted from the timeseries by averaging the minimum/maximum amplitude from each QBO cycle, respectively.

The sum of the squares of the amplitudes between the Fourier harmonics that correspond to the minimum and maximum QBO period over the square root of the field variance at each gridpoint gives the latitude-height QBO structure (Figure 2, middle

panel). From this, the vertical profile at the equator is taken (Figure 2, top panel), which gives the vertical extent of the QBO, defined as the full depth at half maximum, as well as the lowermost depth of the QBO (the lowermost level affected), which is defined as the level of 10% of the maximum amplitude. From the horizontal cross section at the height of the QBO maximum (Figure 2, bottom panel), the latitudinal extent (width) is defined by the full width at half maximum of a fitting Gaussian. The QBO Fourier amplitude is identified as the maximum amplitude, following Pascoe et al. (2005).

One additional metric was derived by looking at the development of the profile of equatorial zonal wind: the descent rate of the shear zones. Figure 3 illustrates the procedure: at each point in time, the height of the sign change ($\bar{u} = 0$) of the wind profile is found by linear interpolation between two subsequent $\bar{u}$ values of opposite sign. The difference between the heights $\Delta h = h_{t+1} - h_t$, divided by the time resolution $\Delta t = 1$ month gives the descent rate. The mean of the descent rates between 10 and 70hPa is calculated separately for the two shear zones.

The metrics for the temperature field are derived in an analogous way from the Fourier spectrum of the $T$ timeseries. QBO temperature characteristics include the Fourier amplitude, height of maximum, latitudinal and vertical extent.





## 4    Model performance

Tables 3 and 4 list the characteristic metrics for all CMIP5 and CCMVal-2 models that have an internally generated QBO, for comparison with the reanalysis datasets and FUB observations (where possible). Table 5 compares the multi-model mean and to the mean of the three most recent reanalyses. Figure 4 shows the multi-model and -reanalysis mean latitude-height QBO

amplitude.

The progress of QBO simulation in GCMs is noticeable: Most models represent the wind amplitude well compared to reanalyses and observations for both easterly and westerly shear zones. Apart from 3 models (CMCC-CMS, UMUKCA-METO and -UCAM), the range of QBO periods is realistic (Table 3), with the multi-model mean not being significantly different from observations and reanalysis mean (Table 5).

A common model bias is a QBO that peaks slightly too high and does not descend low enough as seen in Figure 4. This indicates that the whole QBO structure is shifted slightly upwards. Even at the height of the maximum QBO amplitude, which itself is realistic, the simulated QBOs are too narrow in their latitudinal extent (Table 5). The reanalyses that resolve the atmosphere up to at least 1hPa (all except NCEP1/NCEP2) consistently show the maximum QBO at 20hPa, which is broadly in agreement with the FUB observations, given that the 15hPa level is not included in the reanalyses.

In the temperature field, half of the models peak at a realistic height (20-30hPa), whereas the other half peaks too high ($\sim$5 hPa) which leads on average to an elongated structure in height for the QBO temperature amplitude. Again, the difference between the model and the reanalysis mean shows a shift of the QBO structure upwards. Additionally, there is a slight overestimation of the QBO temperature amplitude at subtropical latitudes (15°-30°) in the models. Exclusion of models with obvious shortcomings in QBO modelling as seen by unrealistic periods does not significantly improve these biases (Table 5).

There is a slight asymmetry in the descent rates of easterly and westerly shear zones in models, but it is not as pronounced as in the observations/reanalyses, where the westerlies descend about twice as fast as the easterlies. Figure 5 shows the easterly and westerly descent rates for each model and reanalysis dataset as well as the mutli-model/reanalysis mean and standard deviations. Even the model with the fastest descending westerlies still has a slower descent rate than the observations and the slowest reanalysis dataset. Most of the models have comparable westerly and easterly descent rates, with UMSLIMCAT even

reversing the asymmetry towards faster easterlies. While within reanalyses and the FUB observations, the standard deviation in the easterly descent rate is usually slightly larger than in the westerly descent rate, the inter-model/-reanalysis discrepancy is higher for descending westerlies. Models show similar standard deviations for both westerly and easterly descent rate, which can also be seen in a more uniform descent of both shear zones and less prominent stalling features compared to the observations (Figure 1).

Figure 6 shows the timing of the phase change at the height of the maximum QBO amplitude. For both west to east and east to west transitions, there is a seasonal modulation in the models with more changes occuring in boreal spring and autumn, but this modulation is not as prominent as in the FUB observations, where west to east transitions are favoured in May and November and east to west transitions are slightly more common in November. Reanalyses favour west to east transitions in October and east to west transitions in December. However, with only 29 FUB observational cycles and 39 (3x13) in total in the reanalyses





to compare, no conclusive statement about the significance of the difference between models and reanalyses/observations can be made. It is, however, intriguing that the distributions of the west-east and east-west transitions look similar in the models, but not in the observations/reanalyses.

## 5   Discussion and Conclusion

The representation of the stratospheric zonal mean wind and temperature fields in eleven models and seven reanalysis datasets was assessed in this paper. It is a positive development that an increasing number of global climate models resolve the stratosphere well enough to show an oscillation in zonal mean zonal wind that resembles the observed QBO.

A set of metrics to characterise the quality of these simulations is established and the model performance is evaluated using reanalyses and the FUB observational radiosonde dataset as reference. Some typical features of the QBO are well represented,

such as the asymmetry in easterly/westerly amplitude, the latitudinal confinement around the equator and the vertical extent. Apart from three models, the mean period and its variability is captured well. However, the QBO in all models is shifted upwards in height compared to reanalyses and narrows in latitude in the lower stratosphere stronger than the reanalyses (Figure 4). Even at the height of the maximum QBO, the modelled QBOs are too narrow, which suggests that the Coriolis parameter may not be the only factor influencing the width as suggested by Haynes (1998). The parametrization of the gravity wave

sources or the width of the inter-tropical convergence zone might play a role as well. However, the disagreement between reanalyses is also greatest at low latitudes as noted by Kawatani et al. (2016), a finding they explain by the small equatorial Coriolis parameter and sparse observations.

The discrepancy between the timing of phase transitions in the reanalyses and observations (Figure 6) was also pointed out by Kawatani et al. (2016). Model behaviour differs even more from the observations, with similar phase transition distribution for

both east-west and west-east transitions. Kawatani et al. (2016) suggest that weak forcing by resolved waves contributes to the bias in reanalysis, a mechanism that might also lead to the discrepancy in models. Furthermore, parametrized gravity waves in the models used in this study are not coupled the main generation processes in the atmosphere, such as tropical convection, which might explain why the annual variation in phase transitions is not as prominent as in the observations.

Insufficient wave forcing might also be responsible for the lack of difference between easterly and westerly descent rates.

In observations, westerlies descend on average about twice as fast as easterlies, whereas in models the difference in rates is not significant, with the westerlies descending too slowly. The standard deviation of the multi-model and -reanalysis mean is clearly higher for westerly than for easterly descent rates, a result that also points towards disagreement in the underlying westerly forcing.

In summary, there has been substantial improvement in simulating the tropical stratosphere in global climate models, with

QBO-like oscillations being represented in a growing number of models. The characteristic metrics defined here present the possibility of quickly assessing the quality of a simulation. With improving model resolution and (subsequently) the representation of wave forcing, GCMs are very likely to simulate a more realistic QBO.





## 6    Data availability

CMIP5 and CCMVal-2 climate model data was downloaded from the British Atmospheric Data Centre (BADC), HadGEM1
data can be obtained from SMO. For reanalysis data please contact Masatomo Fujiwara, who prepared it for the SPARC
reanalysis intercomparison project (S-RIP), or the respective centre as listed here http://s-rip.ees.hokudai.ac.jp/resources/links.

5    html.

*Acknowledgements.*    We acknowledge the World Climate Research Programme's Working Group on Coupled Modelling, which is responsible for CMIP, and we thank the climate modeling groups for producing and making available their model output. We thank the modelling groups of the SPARC Chemistry-Climate Model Validation (CCMVal-2) for producing and making available their model output. We thank the British Atmospheric Data Centre (BADC) for providing access to CMIP5 and CCMVal-2 data and the SPARC reanalysis intercomparison

10    project (S-RIP) for the reanalysis datasets.

SMO and LJG would like to to acknowledge funding from the National Environment Research Council projects QBOnet (NE/M005828/1)
and GOTHAM (NE/P006779/1). The authors would also like to thank Adam Scaife, Andrew Bushell, Jeff Knight and Martin Andrews for
their valuable contributions to early discussions helping to motivate this study. We thank James Anstey, Kevin Hamilton, Thomas Krismer,
John McCormack, Marv Geller and Tiehan Zou for early discussions on and contributions to the SPARC-QBOi questionnaire.





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





| Model | Reference | Resolution | GW scheme | Length |
|---|---|---|---|---|
| HadGEM1 | Osprey et al. (2010) | N96 L60 | W &M | 50 years |
| | Hardiman et al. (2010) | | | |
| | Bushell et al. (2010) | | | |
| MIROC-ESM-CHEM | Watanabe et al. (2011), | T42 L68 | Hines | 156 years |
| | Watanabe and Kawatani (2012) | | | |
| MPI-ESM-MR | Schmidt et al. (2013), | T63 L95 | Hines | 156 years |
| | Krismer and Giorgetta (2014) | | | |
| HadGEM2-CC | Osprey et al. (2013) | 1.25° x 1.875° L60 | W &M | 374 years |
| | Hardiman et al. (2012) | | | |
| CMCC-CMS | Manzini et al. (2006), | T63 L95 | Hines | 156 years |
| | Giorgetta et al. (2006) | | | |
| EMAC | Jöckel et al. (2006) | T42 L90 | Hines | 41 years |
| MRI | Shibata and Deushi (2008a), | T42 L68 | Hines | 47 years |
| | Shibata and Deushi (2008b) | | | |
| UMSLIMCAT | Tian and Chipperfield (2005) | 2.50° x 3.75° L64 | W &M | 55 years |
| UMUKCA-METO | Morgenstern et al. (2009) | 2.50° x 3.75° CP60 | W &M | 47 years |
| UMUKCA-UCAM | Morgenstern et al. (2009) | 2.50° x 3.75° CP60 | W &M | 45 years |

**Table 1.** Climate models used in the study. HadGEM1 was part of CMIP3, MIROC-ESM-CHEM, MPI-ESM-MR and HadGEM2-CC were part of CMIP5, the rest are CCMVal-2 models. CMIP5 models are runs with a coupled ocean, HadGEM1 and the CCMVal-2 models are atmosphere only runs. The gravity wave (GW) parametrisation schemes are based on either Warner and McIntyre (2001) (W & M) or Hines (1997a, b) (Hines).





| Reanalysis | Reference | Resolution of forecast model |
|---|---|---|
| ERA40 | Uppala et al. (2005) | $T_L$ 159 and N80 reduced Gaussian, L60 |
| ERA Interim | Uppala et al. (2005) | $T_L$ 255 and N128 reduced Gaussian, L60 |
| MERRA | Rienecker et al. (2011) | 0.66° lon x 0.5° lat; 72 sigma levels |
| JRA25 | Onogi et al. (2007) | T106 L40 |
| JRA55 | Ebita et al. (2001) | $T_L$ 319 L60 |
| CFSR | Saha et al. (2010) | T382 L64 |
| NCEP1 | Kalnay et al. (1996) | T62 L28 |
| | Kistler et al. (2001) | |
| NCEP2 | Kanamitsu et al. (2002) | T62 L28 |

**Table 2.** Reanalysis datasets used in the study. The period is 1979-2009 for all reanalyses except ERA40, which covers 1958-2001.

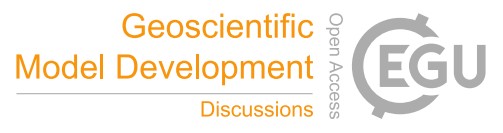
| Model/Reanalysis | Height of maximum (hPa) | Fourier amplitude (m/s) | Latitudinal extent (°) | Vertical extent (km) | Lowest level (hPa) | Period (months) | | | Amplitude (m/s) | | Descent rate (km/mth) | |
|---|---|---|---|---|---|---|---|---|---|---|---|---|
| | | | | | | Min | Max | Mean | East | West | East | West |
| HadGEM1 | 10.0 | 14.7 | 17.9 | 18.1 | 78.4 | 26.2 | 41.3 | 33.8 | -26.8 | 13.0 | 0.7 | 0.7 |
| HadGEM2-CC | 10.0 | 18.6 | 18.7 | 16.7 | 83.5 | 21.1 | 31.2 | 26.5 | -34.8 | 21.4 | 0.8 | 1.1 |
| MIROC-ESM-CHEM | 15.0 | 16.0 | 19.6 | 17.9 | 76.4 | 22.3 | 31.8 | 26.1 | -34.7 | 14.4 | 0.7 | 0.9 |
| MPI-ESM-MR | 10.0 | 22.1 | 20.2 | 18.3 | 84.3 | 23.5 | 57.0 | 30.3 | -38.1 | 26.1 | 0.6 | 0.9 |
| CMCC-CMS | 10.0 | 14.4 | 18.8 | 19.0 | 68.2 | 22.6 | 153.5 | 47.9 | -23.4 | 23.1 | 0.4 | 0.6 |
| EMAC | 15.0 | 13.2 | 21.0 | 18.5 | 75.7 | 24.1 | 35.0 | 28.4 | -29.0 | 14.3 | 0.7 | 1.0 |
| MRI | 20.0 | 13.3 | 20.6 | 21.5 | 80.6 | 25.3 | 38.3 | 28.3 | -27.4 | 12.5 | 0.6 | 1.0 |
| UMSLIMCAT | 10.0 | 15.2 | 18.8 | 22.7 | 76.9 | 28.5 | 35.1 | 31.2 | -30.4 | 19.2 | 0.8 | 0.6 |
| UMUKCA-METO | 10.0 | 11.1 | 18.2 | 15.6 | 80.0 | 53.2 | 62.4 | 56.8 | -20.1 | 11.4 | 0.2 | 0.5 |
| UMUKCA-UCAM | 10.0 | 12.9 | 18.2 | 16.7 | 86.6 | 36.0 | 66.3 | 49.5 | -22.7 | 13.4 | 0.3 | 0.6 |
| ERA 40 | 20.0 | 14.6 | 21.7 | 17.3 | 88.6 | 19.6 | 34.7 | 28.3 | -35.6 | 13.4 | 0.7 | 1.3 |
| ERA-Interim | 20.0 | 14.6 | 20.9 | 15.1 | 85.7 | 22.9 | 34.7 | 28.0 | -32.0 | 14.8 | 0.8 | 1.3 |
| MERRA | 20.0 | 15.1 | 20.8 | 17.7 | 88.8 | 22.5 | 34.8 | 28.0 | -34.6 | 16.4 | 0.7 | 1.2 |
| JRA25 | 20.0 | 14.1 | 20.9 | 19.4 | 87.3 | 22.9 | 35.2 | 28.0 | -35.4 | 16.4 | 0.7 | 1.6 |
| JRA55 | 20.0 | 14.5 | 21.3 | 17.7 | 88.3 | 22.9 | 35.6 | 28.0 | -35.6 | 16.1 | 0.7 | 1.4 |
| CFSR | 20.0 | 14.0 | 21.4 | 16.5 | 82.7 | 23.2 | 34.8 | 27.9 | -34.9 | 14.7 | 0.7 | 1.2 |
| NCEP1 | 20.0 | 11.7 | 20.1 | - | 79.9 | 22.9 | 34.5 | 27.9 | -26.2 | 14.1 | 0.8 | 1.5 |
| NCEP2 | 10.0 | 12.1 | 20.6 | - | 81.4 | 22.5 | 35.1 | 27.7 | -30.8 | 8.8 | 0.8 | 1.7 |
| Observations | 15.0 | 15.7 | - | - | - | 20.4 | 36.9 | 28.2 | -30.8 | 17.1 | 0.7 | 1.2 |

**Table 3.** Calculated QBO metrics for all models and reanalyses. Where possible, the value from the observations is given as well. However, these only consist of one timeseries at the equator (hence no latitudinal information) and are available between 10 and 70hPa for the time 1956-2015, so the depth and vertical extent could not be assessed. The highest level NCEP1 and NCEP2 is 10hPa, which accounts for the missing value in vertical extent.



| Model/Reanalysis | Height of maximum (hPa) | Fourier amplitude (K) | Latitudinal extent (°) | Vertical extent (km) |
|---|---|---|---|---|
| HadGEM1 | 15.0 | 0.7 | 12.9 | 20.0 |
| HadGEM2-CC | 6.0 | 1.0 | 14.4 | 19.2 |
| MIROC-ESM-CHEM | 7.0 | 1.4 | 13.8 | 16.3 |
| MPI-ESM-MR | 5.0 | 1.7 | 15.2 | 20.8 |
| CMCC-CMS | 5.0 | 1.1 | 16.0 | 22.9 |
| EMAC | 20.0 | 1.2 | 15.7 | 17.7 |
| MRI | 30.0 | 0.9 | 15.2 | 19.6 |
| UMSLIMCAT | 20.0 | 1.0 | 13.2 | 21.3 |
| UMUKCA-METO | 30.0 | 0.7 | 12.8 | 19.6 |
| UMUKCA-UCAM | 30.0 | 0.8 | 13.6 | 18.5 |
| ERA 40 | 30.0 | 1.3 | 16.2 | 14.2 |
| ERA-Interim | 30.0 | 1.3 | 16.8 | 14.9 |
| MERRA | 30.0 | 1.3 | 16.8 | 14.8 |
| JRA25 | 30.0 | 1.1 | 15.8 | 17.4 |
| JRA55 | 30.0 | 1.3 | 16.9 | 13.7 |
| CFSR | 20.0 | 1.2 | 17.4 | 15.2 |
| NCEP1 | 30.0 | 0.8 | 15.3 | - |
| NCEP2 | 20.0 | 0.8 | 27.7 | - |

**Table 4.** Characteristic QBO metrics calculated from the zonal mean temperature. Values for models and reanalyses are listed; there is no comparable observational dataset.





| *ZM Zonal Wind* | Model mean | Model mean* | Reanalysis mean |
|---|---|---|---|
| Height of maximum (hPa) | $12.0 \pm 3.5$ | $13.3 \pm 4.1$ | $20.0 \pm 0.0$ |
| Fourier amplitude (m/s) | $15.1 \pm 3.2$ | $16.3 \pm 3.5$ | $14.8 \pm 0.3$ |
| Latitudinal extent (°) | $19.2 \pm 1.1$ | $19.7 \pm 1.2$ | $21.0 \pm 0.3$ |
| Vertical extent (km) | $18.5 \pm 2.2$ | $18.5 \pm 1.6$ | $16.8 \pm 1.5$ |
| Depth (hPa) | $79.1 \pm 5.2$ | $79.8 \pm 3.6$ | $87.6 \pm 1.7$ |
| Mean Period (months) | $35.9 \pm 11.2$ | $28.9 \pm 2.8$ | $28.0 \pm 0.0$ |
| Amplitude Easterly | $-28.7 \pm 5.8$ | $-31.8 \pm 4.7$ | $-34.0 \pm 1.9$ |
| Amplitude Westerly | $16.9 \pm 5.2$ | $17.0 \pm 5.5$ | $15.8 \pm 0.8$ |
| Descent rate Easterly | $0.6 \pm 0.2$ | $0.7 \pm 0.1$ | $0.7 \pm 0.0$ |
| Descent rate Westerly | $0.8 \pm 0.2$ | $0.9 \pm 0.1$ | $1.3 \pm 0.1$ |
| *ZM Temperature* | | | |
| Height of maximum (hPa) | $16.8 \pm 10.7$ | $13.8 \pm 9.9$ | $30.0 \pm 0.0$ |
| Fourier amplitude (m/s) | $1.1 \pm 0.3$ | $1.2 \pm 0.4$ | $1.3 \pm 0.0$ |
| Latitudinal extent (°) | $14.3 \pm 1.2$ | $14.5 \pm 1.0$ | $16.9 \pm 0.0$ |
| Vertical extent (km) | $19.6 \pm 1.9$ | $18.9 \pm 1.6$ | $14.5 \pm 0.7$ |

**Table 5.** Characteristic QBO metrics in reanalyses and models. Values are means and standard deviations of the metrics in Tables 3 and 4. The multi-model mean was calculated from all models (* excluding CMCC-CMS and both UMUKCA models for obvious shortcomings in QBO modelling (Figure 1)), the reanalysis mean from the most recent datasets, namely ERA-Interim, MERRA and JRA55.



**Figure 1.** Equatorial zonal mean zonal wind time-height series from models and the ERA-Interim reanalysis, 1980-2000. Easterly shear zones are blue, westerly shear zones red. The zero wind line is shown in black. The observational dataset from Freie Universität Berlin (2015) is shown on the bottom right for levels 10-70hPa.





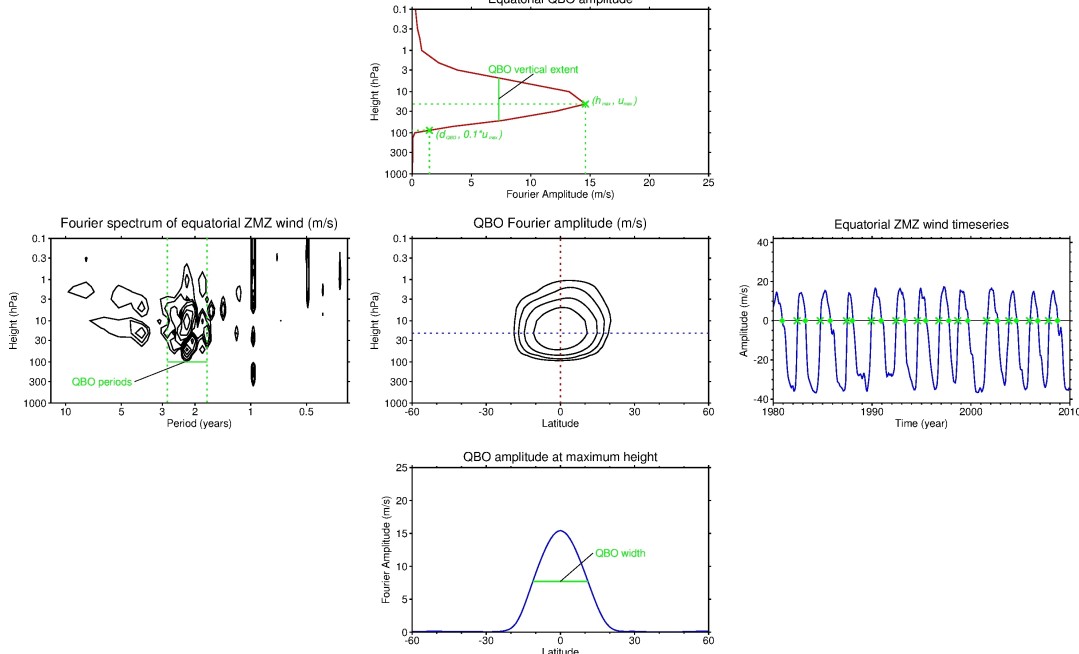

**Figure 2.** Derivation of QBO $\bar{u}$ characteristic metrics, exemplified with the reanalyses mean:

Middle row: Mean Fourier spectrum (left) of equatorial zonal mean zonal wind. Contours are drawn at 1, 2, 4, 8 and 16 m/s. The Fourier harmonics around 2 years are averaged to give the latitude-height QBO amplitude (middle, same contours). From the $\bar{u}$ timeseries at $h_{max}$ (right), the period of each single QBO cycle is calculated and the easterly/westerly amplitudes are identified.

From the latitude-height QBO structure, a cross section at the equator (red) is taken to derive the QBO height profile (upper) and one at 20 hPa (blue) for the latitude profile (lower). From the height profile, the vertical extent, the depth $d_{QBO}$ as well as the maximum Fourier amplitude ($u_{max}$) can be identified. The latitude cross section at $h_{max}$ serves to define the latitudinal extent of the QBO.





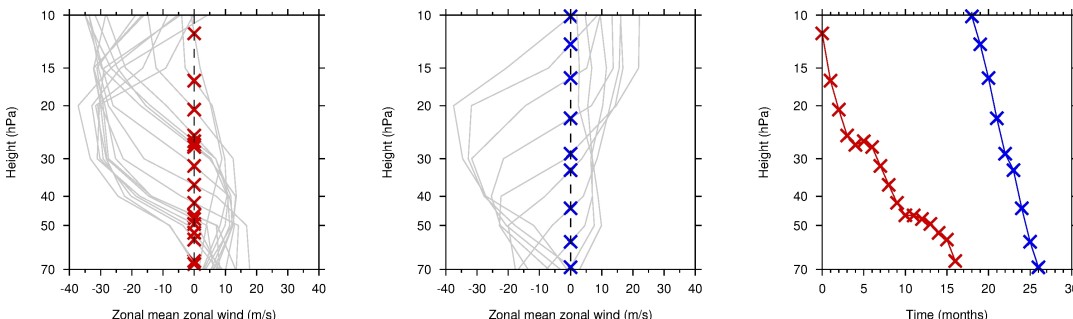

**Figure 3.** Equatorial $\bar{u}$ profiles in consecutive months for a descending easterly (left) and westerly (middle) shear zone from the FU Berlin observations (1964-1966 cycle). The heights of phase change in each month are shown in red/blue and are displayed in the right panel.





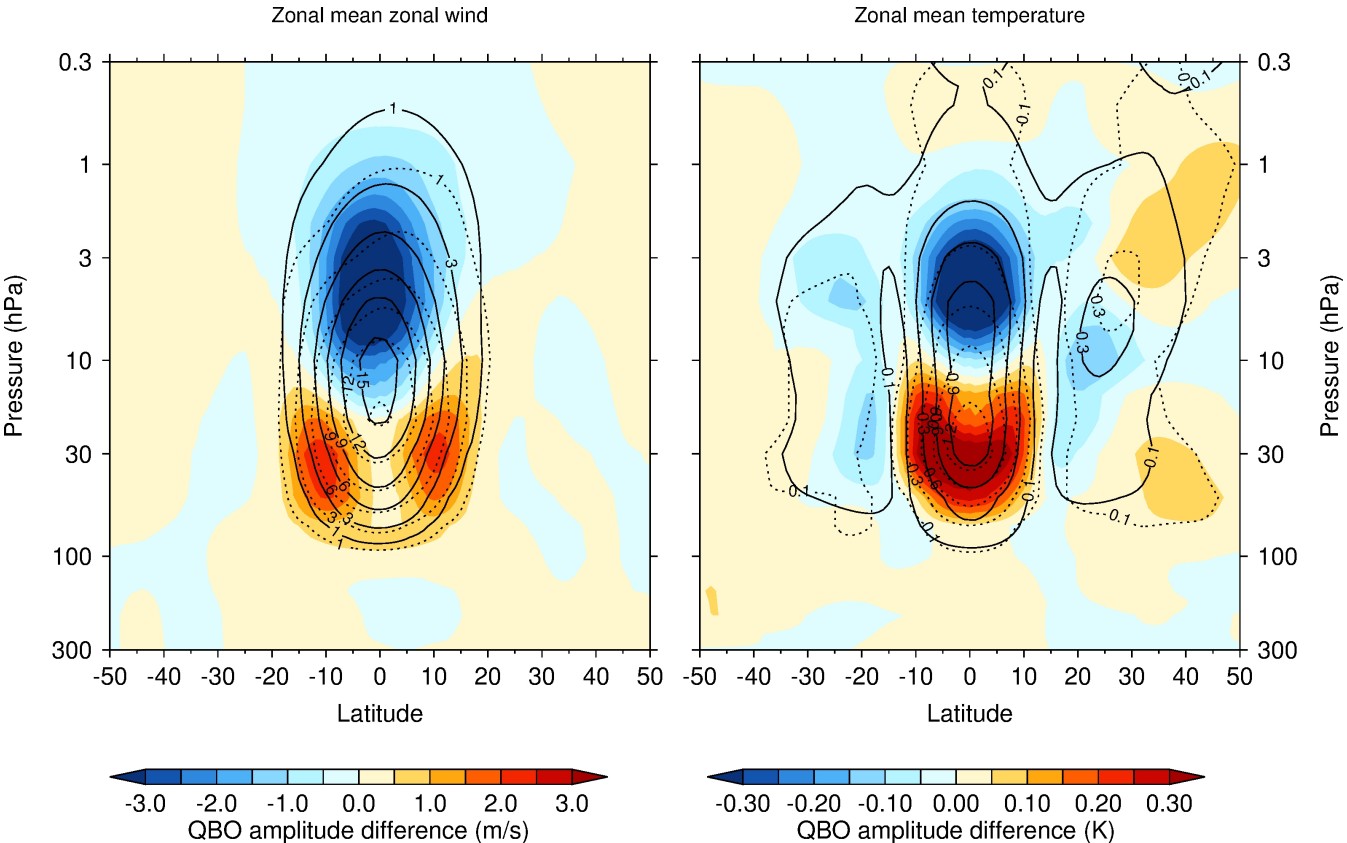

**Figure 4.** Comparison of QBO amplitudes in $u$ (left) and $T$ (right) from models (solid contours) and reanalyses (dotted contours). The colours show the difference reanalyses-models with blue depicting an overestimation by models and red an underestimation.

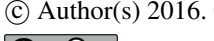



**Figure 5.** QBO easterly and westerly descent rates in models and reanalyses. The symbols (diamonds for models, circles for reanalyses and triangle for observations) show the mean and standard deviation within each dataset. The filled symbols contribute to the model/reanalysis mean as shown with the black diamond/circle. The dotted line represents equal descent rates for both shear zones as orientation.





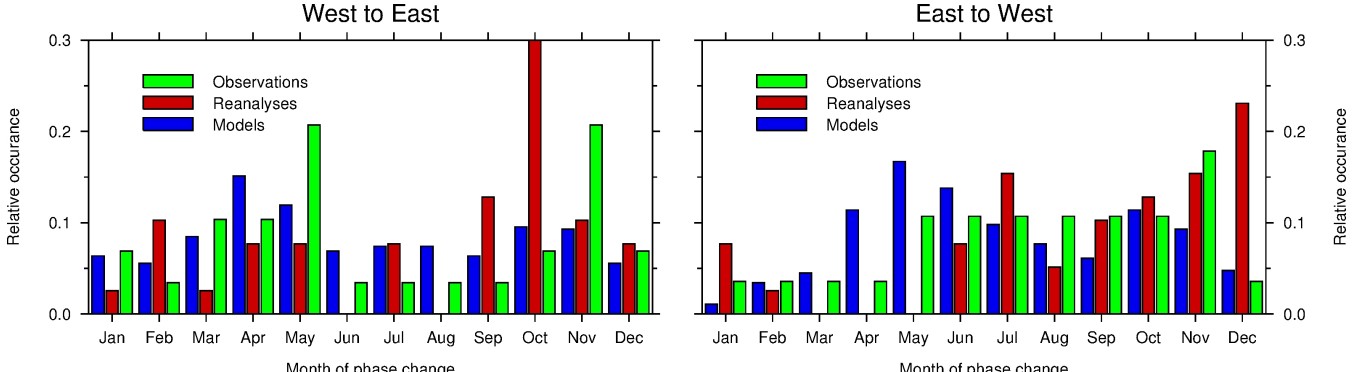

**Figure 6.** Timing of phase change in models (blue, excluding CMCC-CMS and both UMUKCA models), FUB observations (green) and reanalyses (red). There are 407/29/39 west-east changes (distribuation of relative occurence in left panel) and 411/28/39 east-west changes taken into account for models/observations/reanalyses.