# Peer review of "Defining metrics of the Quasi-Biennial Oscillation in global climate models"

_Geoscientific Model Development, 2016_

## Referee Comment (RC1) · Anonymous Referee #1 · 4 Jan 2017

The Quasi-Biennial Oscillation is one of the most important modes of variability in the atmosphere and it is to an increasing extent included in climate models and CCMs. The present paper defines a set of metrics for the QBO and compares these metrics for a set of models and reanalyses.

The subject is important and the paper is well written. However I have a couple of major considerations that the authors should consider before I can recommend that the paper is accepted.

Major comments:

1) I miss some motivations for the chosen metrics and the way they are defined.

For example, some metrics are defined from the Fourier filtered time-series while others seems to be defined from the raw zonal mean zonal wind. What would the difference be if the metrics were calculated from the real data without the Fourier filtering? Why are the mean period defined from calculating zero-crossings in the raw zonal mean zonal wind and not from the spectrum?

In many studies a filtering based on the leading principal components are used (e.g., Wallace 1993, JAS 50, 1751-1762) making it possible to obtain a well defined a phase-speed. This possibility is not even mentioned in the paper.

I also wonder why there is no metric related to the wave-forcing of the QBO included.

The metrics could also be somewhat more detailed described in the text. Even when the caption to Fig. 2 is included the definitions are very densely described. For example, how is the cut-off frequencies of the QBO in the spectrum around two years actually determined?

2) The second half of the paper deals with "model performance" and metrics calculated for the models are compared to those of observations. But there is almost no attempt to address the statistical uncertainty (in e.g. Table 3). Given the relatively few QBO events on record this is an important part of the analysis both for the comparisons in this paper and in general.

The monthly QBO data can because of the oscillatory nature not be modelled by simple processes such as white noise or AR1 models. Christiansen 2010 (JCLIM, 23, 3953-3966) demonstrates one way to overcome this with a Monte Carlo method.

Anyway, the authors should address this problem and provide uncertainty intervals for the numbers in the tables.

Minor comments:

Lines 11, 16: What is meant by "easterly/westerly shear zones"?. Here it seems just to be the easterly/westerly phases the zonal mean wind.

[Figure]

Fig. 3 and 5: Are the profiles in Fig. 3 for one single QBO event? And are the mean and standard deviations shown in Fig.5 then taken over all QBO events?

Perhaps more details about the models could be included in Table 1 regarding the parameterizations of the orographic/non-orographic gravity waves.

Table 2: Is "mth" in the unit for decent rates the same as "months"?

---

## Referee Comment (RC2) · Anonymous Referee #2 · 19 Jan 2017

This study evaluates the QBO as represented in recent climate models, using the small number of CMIP5 and CCMVal models that represent the QBO. The main point of the study is establish the set of metrics that are used here to characterize the QBO, and the authors advocate that these metrics be used in future multi-model comparisons such as those expected from the SPARC QBOi activity. An interesting finding is that the models, on average, have QBOs that are shifted upward and are meridionally too narrow, in comparison to reanalyses.

The proposed metrics are potentially a timely and useful contribution. However I have some issues with the way in which they are presented:

1. The method for calculating the metrics should be presented in a crystal-clear, algorithmic fashion. Since the point is for future studies to repeat these calculations on

different models and/or reanalyses, it needs to be very clear how to do this. I don't think the description of the calculations is sufficiently clear in the present draft. Please see detailed comments in the line-by-line remarks, below.

2. The metrics are presented as-is, with virtually nothing being said on why these choices were made and not others. For example, other ways of defining the QBO amplitude have appeared in the literature, such as Baldwin and Gray 2005. It would be useful for the authors to make the case as to why they settled on these particular choices. Otherwise I speculate that later authors might choose different metrics to characterize the QBO, if this paper hasn't convinced them that the choices made here are well founded. For example, why not just simply use the RMS monthly-mean zonal-mean wind amplitude at a set of standard pressure levels as the measure of QBO amplitude?

3. The metrics in Tables 3 and 4 have no uncertainty estimates associated with them and I see no reason for that omission. The results are mostly given to three significant figures but there is no sense of how meaningful this precision is. Table 6 does give estimates, associated with the multi-model ensemble spread. But for single models (and reanalyses), shouldn't it be possible to give uncertainties based on the internal variability? That is, the variation between QBO cycles.

Based on these issues, and other detailed comments below, I recommend major revisions.

Some other suggestions:

1. Plots for individual models would be useful as supplemental material. For example you could make Fig. 4 for each model individually.

2. Amplitude of the QBO in temperature at the tropical tropopause would be a useful metric. You would need to define the tropopause, but perhaps even just providing the amplitude at 100 hPa would be a simple and useful way to do it.

3. It might be useful to state, in your discussion section, what interesting properties of the QBO are not captured by these metrics. For example some characterization of the zonal momentum budget would be interesting. I'm not suggesting the paper needs to include that, but it would good to state why it doesn't. Data not available in the CMIP5 archive? A desire for simplicity?

4. Histograms showing the distribution of QBO period in each model could be useful (a multi-panel plot, one panel per model). Fig 6 is useful, but the models might show interesting variations amongst themselves. It would show whether some models tend to be more synchronized with the annual cycle than others.

Comments/suggestions by page and line number:

6: "QBO like" –> "QBO-like"

12: "ERA-Interim" –> "ERA-Interim reanalysis"

16: insert "known" after "repeatable"

16: ", beyond" –> " outside of"

17: From Osprey et al and Newman et al I think we've learned that whether the QBO remains as regular in present-day climate is also an outstanding question!

2-3: Join these two paragraphs together, since they both describe the basic QBO theory.

7: insert "reasonably" before "realistic"

8: "Follow on" –> "Follow-on"

8: delete "signal"

9-10: "depends, amongst others," –> "is affected by"

14: Four out of thirty sounds pretty bad, but on the other hand many of these models might have poor stratospheres in general, with model lids below the stratopause. Do you have an estimate of how many of the CMIP5 models can be regarded as "stratosphere-resolving" but still don't produce a QBO?

17: "aims" –> "aim", "are" –> "is"

20: "An additional purpose is to provide" –> "The purpose is to provide"

21: "the future QBO simulations" –> "new QBO-resolving" (so as not to suggest that only future projections are of interest)

29: "Merra" –> "MERRA"

1-2: Suggest deleting this first sentence, it doesn't really add anything. You might instead start this section by introducing Figure 1, since otherwise the figure is first introduced in parentheses near the end of the paragraph, which is easily missed.

4: "was quickly established": not sure what you're referring to here. A previous project comparing QBOs in different models?

5-6: "a typical oscillation with one constant restoring force": I'm not sure what this means. Perhaps you mean "a single restoring force"? For a simple pendulum, F = -kx (Hooke's Law), so F is not constant (its magnitude and direction change). And "typical" is an odd choice in this context: do you mean in comparison to other atmospheric oscillations? It might be simpler to just say that the QBO period is variable, and then go one to explain (as you do from line 6) what might be the causes of the variable period.

8: "these different aspects" –> "the different aspects of the QBO"

8: "Figure" –> "e.g., Figure"

9: suggest delete ", for example,"

10: add comma after "extent".

11-17: On p. 2 you say, "The aims of this paper are to establish a set of standard metrics that comprehensively characterise the QBO." To be used by subsequent studies, the procedure for calculating these metrics needs to be unambiguous. I suggest you provide here a very clear algorithm (set of steps) that you used to calculate the metrics. Something like Charlton and Polvani 2007 ("A new look at SSWs, Part I"), Sec. 2b, is ideal: a numbered list of clearly described steps. Otherwise the reader has to fish through the text for the details, and it is easy for you to inadvertently omit some details. For example, in the caption of Fig 2 you say, "The Fourier harmonics around 2 years are averaged". You need to define the exact range of periods used. They are indicated by vertical lines in the left middle panel of Fig 2, but numbers need to be given so that the diagnostic is reproducible by others. It would also be worth mentioning that this introduces a dependence on the QBO period into all subsequent metrics that are based on the averaged Fourier amplitude, depending on the degree to which a given model's QBO period (which is variable) falls within the chosen range.

13: "height" –> "altitude". Similarly in Fig. 2 title of bottom panel.

14: "QBO period" –> "distribution of QBO periods"

16-17: What is the min/max amplitude "from each QBO cycle"? Is it just the min/max wind, or wind shear? If so then remove "amplitude", or otherwise define how amplitude is calculated for a single QBO phase. Also state explicitly whether it's a wind amplitude, or vertical wind shear amplitude, or both, that you're calculating. You say "shear zone", but you're discussing a time series of the wind at a single altitude.

18: I think you mean the sum of squared amplitudes of Fourier harmonics that fall between the min and max QBO periods? State how the min/max QBO periods are

determined: are these assumed values? (see comment for lines 11-17, above). This is potentially misleading because in the previous paragraph you said that the min/max QBO period is determined from the timeseries of u at h_max. But I assume you can't be referring to these periods here because h_max hasn't yet been defined, since you're describing here how you determine the latitude-height structure. So the order of presentation between the previous paragraph and this one is confusing. A clear, algorithmic description of how the metrics are calculated could fix this.

22: "maximum amplitude" –> "maximum"

23: Why is a fitted Gaussian used? Why not just use the latitude-altitude structure itself, as was done for the vertical depth? If a Gaussian is required for some reason (the reason should be stated), is it always a good fit? Does the fit quality vary amongst models? I'm worried that in comparing the values of this metric for different models, if a Gaussian is a good fit for one model but not another then the comparison may be less meaningful.

23-24: "The QBO Fourier amplitude...": this sentence seems out of place here, since you have already referred to the maximum. Also, still unclear what is "maximum amplitude": is it just the maximum? The term "amplitude", here and leading up to this point, seems to be used carelessly. Amplitude is itself a metric, which can be defined in various ways, e.g. RMS amplitude of a time series.

27: "subsequent u values of opposite sign" –> "values of u having opposite sign at adjacent gridpoints" (or similar. "subsequent" seems the wrong word here)

6: "The progress... is noticeable": Do you mean from older to newer models in your set of models? If so, you could refer to Table 1 as indicating the vintages of the different models (by the year of the references given). Or, if you mean with respect to earlier results in the literature, please provide some specific comparisons.

11: insert "on average" after "QBO structure"

11-12: Table 5 shows that the models and reanalysis disagree on h_max, i.e. the model error bars do not overlap the reanalysis value. So it seems incorrect to say that h_max in the models is realistic. This is also clear from Table 3, first column (h_max is 10 hPa for all but three models). The disagreement is consistent with your general result that the QBO in the models is shifted upward with respect to reanalyses.

16: insert " (Figure 4)" after "temperature amplitude"

2-3: Does the timing of phase transitions agree better between obs and reanalyes if you exclude some of the older reanalyses, such as NCEP1/2 and perhaps also JRA-25?

5: In Table 5 I count ten models and eight reanalyses. Also, you assessed the observations (FUB winds).

8: "was established" and "was assessed" (previous paragraph used past tense - be consistent)

11: I'm not sure where you commented on the variability of the QBO period in the models. Table 3 shows the min/max period, but plots of the distribution of periods would be more informative.

12: "narrows" –> "is narrower", and "stronger than" –> "than in"

14: I'm not sure this the correct way to state Haynes (1998)'s result. That paper shows that the QBO width not set by the width of the forcing when the imposed wave forcing is prescribed to have a very wide latitudinal distribution, designed not to impose a latitudinal scale on the QBO. I don't see that it rules out the actual forcing having a latitudinal distribution that might affect the QBO width. You note that the width of the ITCZ and/or imposed gravity wave sources may play a role, and I agree.

22: "coupled" –> "coupled to"

26-27: In Table 5, the standard deviation of descent rates for the models is the same for westerlies and easterlies. Either this statement is wrong or Table 5 is wrong.

31: If you mean that increased resolution leads to better representation of the wave forcing, perhaps change "(subsequently)" to "concomitantly"

Table 1

- According to the text (p. 2), there are four CMIP5 models, not three. I believe CMCC-CMS is also a CMIP5 model, and shares many similarities with MPI-ESM-MR. Please correct the caption.

Table 3

- are confidence intervals for some of these columns appropriate? e.g.mean period.

- why are the descent rates reported with fewer significant figures than the other metrics?

Table 4

- for temperature, lowest level (as in Table 3 for wind) would be a useful metric.

Table 5

- For the reanalysis column, a number of the error values are zero.

- "Values are means and standard deviations of the metrics in Tables 3 and 4" –> "The mean and +/- one standard deviation of the metrics in Tables 3 and 4 are shown."

- "excluding CMCC-CMS and both" –> "excluding both CMCC-CMS and"

- change "Depth" in the table to "Lowest level", to be consistent with Table 3

- why are the min/max periods not included? (all other metrics from Tables 3,4 are included)

Fig 1

- It would be helpful to expand this figure in the vertical (pressure) direction. Right now all the panels look kind of squished.

- Label the middle panel to indicate that h_max is the blue horizontal line.

- The blue and red lines in the middle panel are helpful. It's good how they correspond to the colours of the lines in the top, right, and bottom panels. But the dashed line style makes it easy to miss the colours. Perhaps make these solid lines.

- It would help to add arrows between the panels indicating the algorithm for calculating the metrics. That is, an arrow from the left (Fourier spectrum) pointing at the middle panel (latitude-altitude QBO amplitude), and then arrows from the middle panel point outward at the other three panels.

Fig 4

- Since the filled contours show the model bias (with respect to reanalyses), it would be more conventional to show the model-minus-reanalysis difference.

Fig 6

- This is subjective, but I find it very hard to compare the shapes of the three datasets in this format of plot. You might consider using a six-panel plot to show these results. You could have the phase transition direction as the row and the datasets as the columns (the plots could be narrower with only one dataset shown on each one).
* * *

---

## Referee Comment (RC3) · Anonymous Referee #3 · 27 Jan 2017

The QBO is the primary mode of variability in the Tropical stratosphere. The current paper aims to establish a set of standard metrics that comprehensively characterize the QBO. Subsequently the metrics are applied to 10 global circulation models, observations and reanalysis.

This paper is a very useful contribution, however I have some concerns and hence recommend major revisions.

Major Concerns:

1) The primary goal of this paper is to establish a standard set of metrics that can be used in the future. Ideally, the paper should include codes for calculating the metrics, so they are easily reproducible by other groups – hence point to a website from which such a diagnostic package can be downloaded. At the very least include very clear, step-bystep instructions on how the metrics were calculated should be included (without any ambiguity). The metrics presented here are reasonably well described, however there are lots of details in calculations, especially related to calculating the Fourier spectrum (step 1) which are omitted.

2) The paper somewhat lacks a description of what are the science goals motivating these metrics. The presented metrics seem useful to the general assessment of the representation of the QBO in global models, however they do not address aspects related to studying QBO related phenomena, such as QBO teleconnections for example. Hence, the use of these metrics is somewhat limited.

3) The Fourier analysis is useful in certain respects for the assessment of the QBO (such as hmax, mean period), however from the mean and min/max QBO period values presented in Table 3 it is difficult to assess whether a model is getting the correct period distribution. The periods of the QBO vary between 20 and 35 months, and a simple histogram showing the number of times each period occurs would be more helpful in comparing observations to model output.

4) It would be nice to see all the diagnostics for all the models in the appendix (ie.: Figure 2, Figure 4, Figure 6). The multi-model mean is nice to see and the numerical diagnostics are listed in Table 3, but the figures contain so much more information - it would be nice to see the complete set of metrics for all the models.

5) The metrics do not address the forcings of the QBO: gravity waves, resolved waves, vertical advection. It is possible for the QBO characteristics to be very close to observations, and for the forcing mechanisms to be unrealistic (ie.: lack of contribution from resolved waves, etc). Hence the addition of metrics addressing the momentum driving of the QBO would be a very important metric to add.

Minor Comments:

1) Page 2, Line 23: There is an inconsistency between 'four CMIP5 models, and 5

[Figure]

CCMVAL models' here and Table 1. In Table 1 only 3 models are listed as part of CMIP5: MIROC-ESM-CHEM, MPI-ESM-MR and HadGEM2-CC ; I believe the CMCC-CMS should be included in the list of CMIP5 models in the caption of Table 1.

2) Page 3, Line 3: 'the period of the oscillation. . .' - this should say 'the mean period of the oscillation' - it is well know that the period varies quite a bit as noted further in that paragraph

3) Figure 2 caption: What is hmax ?

4) Page 5, Line 5: 'eleven models' – aren't there only 10 in Table 4?

———————————————

---

## Author Comment (AC1) · 12 Mar 2017

The Quasi-Biennial Oscillation is one of the most important modes of variability in the
atmosphere and it is to an increasing extent included in climate models and CCMs.
The present paper defines a set of metrics for the QBO and compares these metrics
for a set of models and reanalyses.
The subject is important and the paper is well written. However I have a couple of
major considerations that the authors should consider before I can recommend that
the paper is accepted.
Major comments:
1) I miss some motivations for the chosen metrics and the way they are defined.

Added "These metrics were defined to be as simple as possible, yet meaningful in characterising the
QBO morphologically.  For robust and simple assessment of the QBO in models and observations,
this study focusses on the large-scale morphology of the QBO rather than those (small-scale)
dynamical processes involved in maintaining it." (p. 2, l. 18-20)

For example, some metrics are defined from the Fourier filtered time-series while others seems to
be defined from the raw zonal mean zonal wind. What would the difference
be if the metrics were calculated from the real data without the Fourier filtering?

This is not the case. Metrics are either defined from the raw zonal wind or the spectrum. No
filtering is applied. The section describing the metrics definitions has been rewritten for clarification
(p. 3, l. 3 - p. 4, l. 10).

Why are the mean period defined from calculating zero-crossings in the raw zonal
mean zonal wind and not from the spectrum?

Using the zonal wind is a more intuitive and accurate way to define the period and give an error
estimate. Looking at the Fourier spectrum of the ERA-Interim reanalysis (Figure 2, left panel), the
periods calculated from the timeseries compare well to the broad spectral peak.

In many studies a filtering based on the leading principal components are used (e.g.,
Wallace 1993, JAS 50, 1751-1762) making it possible to obtain a well defined a phasespeed.
This possibility is not even mentioned in the paper.

One of the aims was to define metrics as simple and intuitively as possible - PC analysis introduces
an additional analysis step that would need further justification; calculating periods from the first
two principal components for the observations gives the same result as from the wind within the
standard deviation interval ($28.0 \pm 3.6$ vs. $28.2 \pm 4.4$ months). This was added as a footnote in the
metrics description (p. 3)

I also wonder why there is no metric related to the wave-forcing of the QBO included.

The metrics are primarily for  phenomenological assessment and also the necessary data were not
available for most of the models. Added "These metrics were defined to be as simple as possible,
yet meaningful in characterising the QBO morphologically.  For robust and simple assessment of
the QBO in models and observations, this study focusses on the large-scale morphology of the QBO

rather than those (small-scale) dynamical processes involved in maintaining it." (p. 2, l. 18-20) to clarify.

The metrics could also be somewhat more detailed described in the text. Even when the caption to Fig. 2 is included the definitions are very densely described.

Agreed - the definition of metrics has been expanded to an algorithmic description (p. 3, l. 3 - p. 4, l. 10).

For example,
how is the cut-off frequencies of the QBO in the spectrum around two years
actually determined?

Cut-off frequencies are calculated from the minimum and maximum period; description has been changed to clarify (p. 3, l. 22).

2) The second half of the paper deals with "model performance" and metrics calculated for the models are compared to those of observations. But there is almost no attempt to address the statistical uncertainty (in e.g. Table 3). Given the relatively few QBO events on record this is an important part of the analysis both for the comparisons in this paper and in general.

Added paragraph "Error estimation" (p. 4, l. 15-30) and numbers in tables 3 and 4.

The monthly QBO data can because of the oscillatory nature not be modelled by simple processes such as white noise or AR1 models. Christiansen 2010 (JCLIM, 23, 3953-3966) demonstrates one way to overcome this with a Monte Carlo method.

Thank you for the suggestion. The method was included in error estimation for the minimum/maximum period and the Fourier amplitude (p. 4, l. 18-25). Outside the area that is dominated by the QBO (above 10hPa, below 70hPa, further away from the equator) the method unfortunately cannot be used as no clear QBO cycle can be defined.

Anyway, the authors should address this problem and provide uncertainty intervals for the numbers in the tables.

Uncertainties are now included in tables 3 and 4.

Minor comments:
Lines 11, 16: What is meant by "easterly/westerly shear zones"?. Here it seems just to be the easterly/westerly phases the zonal mean wind.
Changed wording to "phase" where appropriate.

Fig. 3 and 5: Are the profiles in Fig. 3 for one single QBO event? Yes. The caption states says that these are from the 1964-1966 cycle. And are the mean
and standard deviations shown in Fig.5 then taken over all QBO events? Yes. In the metrics description, it is defined as "The mean of the descent rates between 10 and 70hPa is calculated separately for the two shear zones as the mean over all values for a descending easterly/westerly." (p. 4, l. 10)

Perhaps more details about the models could be included in Table 1 regarding the parameterizations of the orographic/non-orographic gravity waves. Relevant references are included

in the table.
Table 2: Is "mth" in the unit for decent rates the same as "months"? Changed mth->month

Thank you for your review.

**Anonymous Referee #2**

This study evaluates the QBO as represented in recent climate models, using the small number of CMIP5 and CCMVal models that represent the QBO. The main point of the study is establish the set of metrics that are used here to characterize the QBO, and the authors advocate that these metrics be used in future multi-model comparisons such as those expected from the SPARC QBOi activity. An interesting finding is that the models, on average, have QBOs that are shifted upward and are meridionally too narrow, in comparison to reanalyses.
The proposed metrics are potentially a timely and useful contribution. However I have some issues with the way in which they are presented:
1. The method for calculating the metrics should be presented in a crystal-clear, algorithmic fashion. Since the point is for future studies to repeat these calculations on different models and/or reanalyses, it needs to be very clear how to do this. I don't think the description of the calculations is sufficiently clear in the present draft. Please see detailed comments in the line-by-line remarks, below.
Thank you for this remark. The definition of the metrics has been changed to an algorithmic description.
(p. 3, l. 3 - p. 4, l. 10).

2. The metrics are presented as-is, with virtually nothing being said on why these choices were made and not others. For example, other ways of defining the QBO amplitude have appeared in the literature, such as Baldwin and Gray 2005. It would be useful for the authors to make the case as to why they settled on these particular choices. Otherwise I speculate that later authors might choose different metrics to characterize the QBO, if this paper hasn't convinced them that the choices made here are well founded. For example, why not just simply use the RMS monthly-mean zonalmean wind amplitude at a set of standard pressure levels as the measure of QBO amplitude?

Metrics have been chosen for simplicity and conciseness - having an amplitude definition at a set of levels is more complicated (more numbers) than defining it at one particular level that has been chosen as the level where the QBO is strongest.
Added "These metrics were defined to be as simple as possible, yet meaningful in characterising the QBO morphologically." (p. 2, l. 18-19) to clarify the general approach to choosing the metrics.

3. The metrics in Tables 3 and 4 have no uncertainty estimates associated with them and I see no reason for that omission. The results are mostly given to three significant figures but there is no sense of how meaningful this precision is. Table 6 does give estimates, associated with the multi-model ensemble spread. But for single models (and reanalyses), shouldn't it be possible to give uncertainties based on the internal variability? That is, the variation between QBO cycles.

Added paragraph "Error estimations" (p. 4, l. 15-30) and uncertainties in tables 3 and 4. Unfortunately not all numbers can be derived from variations between cycles; an alternative method has been applied to those (p. 4, l. 18-25).

Based on these issues, and other detailed comments below, I recommend major revisions.
Some other suggestions:
1. Plots for individual models would be useful as supplemental material. For example
you could make Fig. 4 for each model individually.
As requested, the equivalent plots for the individual models were added to supplementary material.

2. Amplitude of the QBO in temperature at the tropical tropopause would be a useful
metric. You would need to define the tropopause, but perhaps even just providing the
amplitude at 100 hPa would be a simple and useful way to do it.
As the models have strong deficiencies to represent the dynamical QBO close enough to the
tropopause, it is questionable how useful this metric would be. Further, temperature amplitudes at
the tropopause are influenced by many other regional factors (e.g. ENSO), making it harder to
attribute a certain variability to the QBO.
To address this comment, the "Depth" measurement for temperature, defined analogous to the
"Depth" from the wind field, has been introduced.

3. It might be useful to state, in your discussion section, what interesting properties of
the QBO are not captured by these metrics. For example some characterization of the
zonal momentum budget would be interesting. I'm not suggesting the paper needs to
include that, but it would good to state why it doesn't. Data not available in the CMIP5
archive? A desire for simplicity? The metrics are defined for simplicity and objectivity and only
deal with the morphology of the QBO. Further, data is not available for all models in the CMIP5
archive and there is little observational reference.
Added " For robust and simple assessment of the QBO in models and observations, this study
focusses on the large-scale morphology of the QBO rather than those (small-scale) dynamical
processes involved in maintaining it." (p. 2, l. 19-20)

4. Histograms showing the distribution of QBO period in each model could be useful
(a multi-panel plot, one panel per model). Fig 6 is useful, but the models might show
interesting variations amongst themselves. It would show whether some models tend
to be more synchronized with the annual cycle than others.
Added to supplementary material.

Comments/suggestions by page and line number:
6: "QBO like" –> "QBO-like" Done.
12: "ERA-Interim" –> "ERA-Interim reanalysis" Done.
16: insert "known" after "repeatable" Done.
16: ", beyond" –> " outside of" Done.
17: From Osprey et al and Newman et al I think we've learned that whether the QBO
remains as regular in present-day climate is also an outstanding question! Of course. Sentence
changed to "Whether or not the QBO remains as regular in the present-day climate and under future
climate change is an outstanding question
".
2-3: Join these two paragraphs together, since they both describe the basic QBO
theory. Done.
7: insert "reasonably" before "realistic" Done.
8: "Follow on" –> "Follow-on" Done.
8: delete "signal" Done.
9-10: "depends, amongst others," –> "is affected by" Done.

14: Four out of thirty sounds pretty bad, but on the other hand many of these models might have poor stratospheres in general, with model lids below the stratopause. Do you have an estimate of how many of the CMIP5 models can be regarded as "stratosphere-resolving" but still don't produce a QBO?

Ten models are stratosphere-resolving and include non-orographic gravity waves (Charlton-Perez, A. J., et al. *(2013), On the lack of stratospheric dynamical variability in low-top versions of the CMIP5 models,* J. Geophys. Res. Atmos., 118, 2494–2505), so should be able to produce a QBO. Added a footnote on page 2.

17: "aims" –> "aim", "are" –> "is" Done.

20: "An additional purpose is to provide" –> "The purpose is to provide" Done.

21: "the future QBO simulations" –> "new QBO-resolving" (so as not to suggest that only future projections are of interest) Done.

29: "Merra" –> "MERRA" Done.

1-2: Suggest deleting this first sentence, it doesn't really add anything. You might instead start this section by introducing Figure 1, since otherwise the figure is first introduced in parentheses near the end of the paragraph, which is easily missed. Agreed and done.

4: "was quickly established": not sure what you're referring to here. A previous project comparing QBOs in different models? Changed to "The most obvious one is the mean period;"

5-6: "a typical oscillation with one constant restoring force": I'm not sure what this means. Perhaps you mean "a single restoring force"? For a simple pendulum, F = -kx (Hooke's Law), so F is not constant (its magnitude and direction change). And "typical" is an odd choice in this context: do you mean in comparison to other atmospheric oscillations? It might be simpler to just say that the QBO period is variable, and then go one to explain (as you do from line 6) what might be the causes of the variable period. Changed to "Furthermore, it is not a classic harmonic oscillation with one single restoring force, which leads to a variety of periods."

8: "these different aspects" –> "the different aspects of the QBO" Done.

8: "Figure" –> "e.g., Figure"

9: suggest delete ", for example,"

10: add comma after "extent".

11-17: On p. 2 you say, "The aims of this paper are to establish a set of standard metrics that comprehensively characterise the QBO." To be used by subsequent studies, the procedure for calculating these metrics needs to be unambiguous. I suggest you provide here a very clear algorithm (set of steps) that you used to calculate the metrics. Something like Charlton and Polvani 2007 ("A new look at SSWs, Part I"), Sec. 2b, is ideal: a numbered list of clearly described steps. Otherwise the reader has to fish through the text for the details, and it is easy for you to inadvertently omit some details. For example, in the caption of Fig 2 you say, "The Fourier harmonics around 2 years are averaged". You need to define the exact range of periods used. They are indicated by vertical lines in the left middle panel of Fig 2, but numbers need to be given so that the diagnostic is reproducible by others. It would also be worth mentioning that this introduces a dependence on the QBO period into all subsequent metrics that are based on the averaged Fourier amplitude, depending on the degree to which a given model's QBO period (which is variable) falls within the chosen range.

The definition of the metrics was clarified (p. 3, l. 5 - p. 4, l. 13); this particular comment is addressed as "The inverse of the minimum/maximum period is taken the upper/lower limit of the QBO Fourier harmonics." (p. 3, l. 23-26)

13: "height" –> "altitude". Similarly in Fig. 2 title of bottom panel. Done.

14: "QBO period" –> "distribution of QBO periods"

16-17: What is the min/max amplitude "from each QBO cycle"? Is it just the min/max wind, or wind shear? If so then remove "amplitude", or otherwise define how amplitude

is calculated for a single QBO phase. Also state explicitly whether it's a wind amplitude, or vertical wind shear amplitude, or both, that you're calculating. You say "shear zone", but you're discussing a time series of the wind at a single altitude.

Description changed to "The amplitude of the easterly/westerly phase in one QBO cycle are defined from the timeseries as the minimum/maximum wind value of a cycle. The values of each cycle are averaged to give the easterly/westerly amplitude". (p. 3, l. 21-11)

18: I think you mean the sum of squared amplitudes of Fourier harmonics that fall between the min and max QBO periods? State how the min/max QBO periods are determined: are these assumed values? (see comment for lines 11-17, above). This is potentially misleading because in the previous paragraph you said that the min/max QBO period is determined from the timeseries of u at h_max. But I assume you can't be referring to these periods here because h_max hasn't yet been defined, since you're describing here how you determine the latitude-height structure. So the order of presentation between the previous paragraph and this one is confusing. A clear, algorithmic description of how the metrics are calculated could fix this.

The definition of the metrics was clarified (p. 3, l. 5 - p. 4, l. 13); this particular comment is addressed as "The inverse of the minimum/maximum period is taken the upper/lower limit of the QBO Fourier harmonics." (p. 3, l. 23-26)

22: "maximum amplitude" –> "maximum" Not applicable anymore due to text changes.

23: Why is a fitted Gaussian used? Why not just use the latitude-altitude structure itself, as was done for the vertical depth? If a Gaussian is required for some reason (the reason should be stated), is it always a good fit? Does the fit quality vary amongst models? I'm worried that in comparing the values of this metric for different models, if a Gaussian is a good fit for one model but not another then the comparison may be less meaningful.

The Gaussian is a good fit for all models and is commonly used in QBO characterisation (e.g. Pascoe et al. (2005)).

23-24: "The QBO Fourier amplitude...": this sentence seems out of place here, since you have already referred to the maximum. Also, still unclear what is "maximum amplitude": is it just the maximum? The term "amplitude", here and leading up to this point, seems to be used carelessly. Amplitude is itself a metric, which can be defined in various ways, e.g. RMS amplitude of a time series. Not applicable anymore due to text changes.

27: "subsequent u values of opposite sign" –> "values of u having opposite sign at adjacent gridpoints" (or similar. "subsequent" seems the wrong word here) Done.

6: "The progress... is noticeable": Do you mean from older to newer models in your set of models? If so, you could refer to Table 1 as indicating the vintages of the different models (by the year of the references given). Or, if you mean with respect to earlier results in the literature, please provide some specific comparisons.

Agreed that this was a vague statement. Changed to "The success of QBO simulation in GCMs is noticeable." (p. 5, l. 6)

11: insert "on average" after "QBO structure" Done.

11-12: Table 5 shows that the models and reanalysis disagree on h_max, i.e. the model error bars do not overlap the reanalysis value. So it seems incorrect to say that h_max in the models is realistic. This is also clear from Table 3, first column (h_max is 10 hPa for all but three models). The disagreement is consistent with your general result that the QBO in the models is shifted upward with respect to reanalyses. Yes, agreed. Deleted the statement about h_max.

16: insert " (Figure 4)" after "temperature amplitude" Done.

2-3: Does the timing of phase transitions agree better between obs and reanalyes if you exclude some of the older reanalyses, such as NCEP1/2 and perhaps also JRA-25?

This was done. Even the more recent reanalyses have problems with phase transition representation (see Kawatani et al (2016)).

5: In Table 5 I count ten models and eight reanalyses. Also, you assessed the observations (FUB winds). Changed.

8: "was established" and "was assessed" (previous paragraph used past tense - be consistent) Done.

11: I'm not sure where you commented on the variability of the QBO period in the models. Table 3 shows the min/max period, but plots of the distribution of periods would be more informative. Thanks for the suggestion. The plots have been included in the supplementary material.

12: "narrows" –> "is narrower", and "stronger than" –> "than in" Done.

14: I'm not sure this the correct way to state Haynes (1998)'s result. That paper shows that the QBO width not set by the width of the forcing when the imposed wave forcing is prescribed to have a very wide latitudinal distribution, designed not to impose a latitudinal scale on the QBO. I don't see that it rules out the actual forcing having a latitudinal distribution that might affect the QBO width. You note that the width of the ITCZ and/or imposed gravity wave sources may play a role, and I agree.

22: "coupled" –> "coupled to" Done.

26-27: In Table 5, the standard deviation of descent rates for the models is the same for westerlies and easterlies. Either this statement is wrong or Table 5 is wrong. Statement removed.

31: If you mean that increased resolution leads to better representation of the wave forcing, perhaps change "(subsequently)" to "concomitantly" Done.

Table 1
- According to the text (p. 2), there are four CMIP5 models, not three. I believe CMCCCMS is also a CMIP5 model, and shares many similarities with MPI-ESM-MR. Please correct the caption. Done; thanks for spotting.

Table 3
- are confidence intervals for some of these columns appropriate? e.g.mean period. Done.
- why are the descent rates reported with fewer significant figures than the other metrics? Changed.

Table 4
- for temperature, lowest level (as in Table 3 for wind) would be a useful metric. Agreed. Included now.

Table 5
- For the reanalysis column, a number of the error values are zero.
- "Values are means and standard deviations of the metrics in Tables 3 and 4" –> "The mean and +/- one standard deviation of the metrics in Tables 3 and 4 are shown." Done.
- "excluding CMCC-CMS and both" –> "excluding both CMCC-CMS and" Both refers to the two UMUKCA models (-UCAM and -METO), so the word order should be correct.
- change "Depth" in the table to "Lowest level", to be consistent with Table 3 Done.
- why are the min/max periods not included? (all other metrics from Tables 3,4 are included) Included now.

Fig 1
- It would be helpful to expand this figure in the vertical (pressure) direction. Right now all the panels look kind of squished. Changed the panel formats.
- Label the middle panel to indicate that h_max is the blue horizontal line. Done.
- The blue and red lines in the middle panel are helpful. It's good how they correspond to the colours of the lines in the top, right, and bottom panels. But the dashed line style makes it easy to miss the colours. Perhaps make these solid lines. Done.
- It would help to add arrows between the panels indicating the algorithm for calculating the metrics. That is, an arrow from the left (Fourier spectrum) pointing at the middle panel (latitude-altitude QBO amplitude), and then arrows from the middle panel point outward at the other three panels.

This is a reasonable suggestion. However, the authors feel that this would make the figure even busier and therefore kept the old format.

Fig 4
- Since the filled contours show the model bias (with respect to reanalyses), it would be more conventional to show the model-minus-reanalysis difference.
Plot/Description changed accordingly.

Fig 6
- This is subjective, but I find it very hard to compare the shapes of the three datasets in this format of plot. You might consider using a six-panel plot to show these results. You could have the phase transition direction as the row and the datasets as the columns (the plots could be narrower with only one dataset shown on each one).
The authors agreed on keeping the current presentation, which admittedly is dense, but therefore needs less space.

Thank you for your valuable comments.

**Anonymous Referee #3**

The QBO is the primary mode of variability in the Tropical stratosphere. The current paper aims to establish a set of standard metrics that comprehensively characterize the QBO. Subsequently the metrics are applied to 10 global circulation models, observations and reanalysis.
This paper is a very useful contribution, however I have some concerns and hence recommend major revisions.

Major Concerns:
1) The primary goal of this paper is to establish a standard set of metrics that can be used in the future. Ideally, the paper should include codes for calculating the metrics, so they are easily reproducible by other groups – hence point to a website from which such a diagnostic package can be downloaded. At the very least include very clear, step-by-step instructions on how the metrics were calculated should be included (without any ambiguity). The metrics presented here are reasonably well described, however there are lots of details in calculations, especially related to calculating the Fourier spectrum (step 1) which are omitted.
Definition of metrics was expanded/changed to an algorithmic description.(p. 3, l. 5 - p. 4, l. 13)

2) The paper somewhat lacks a description of what are the science goals motivating these metrics. The presented metrics seem useful to the general assessment of the representation of the QBO in global models, however they do not address aspects related to studying QBO related phenomena, such as QBO teleconnections for example. Hence, the use of these metrics is somewhat limited.
The purpose is to provide a phenomenological description of the QBO. They can be used in conjunction with teleconnection metrics (which, however, are well beyond the scope of this paper) to assess which QBO characteristics are more relevant for the interactions.
Added "These metrics were defined to be as simple as possible, yet meaningful in characterising the QBO morphologically.  For robust and simple assessment of the QBO in models and observations, this study focusses on the large-scale morphology of the QBO rather than those (small-scale) dynamical processes involved in maintaining it." (p. 2, l. 18-20)

3) The Fourier analysis is useful in certain respects for the assessment of the QBO (such as hmax, mean period), however from the mean and min/max QBO period values presented in Table 3 it is difficult to assess whether a model is getting the correct period

distribution. The periods of the QBO vary between 20 and 35 months, and a simple histogram showing the number of times each period occurs would be more helpful in comparing observations to model output.
A corresponding figure was added to supplementary material.

4) It would be nice to see all the diagnostics for all the models in the appendix (ie.: Figure 2, Figure 4, Figure 6). The multi-model mean is nice to see and the numerical diagnostics are listed in Table 3, but the figures contain so much more information - it would be nice to see the complete set of metrics for all the models.
Figure 4 for models and period distribution added to supplementary material.

5) The metrics do not address the forcings of the QBO: gravity waves, resolved waves, vertical advection. It is possible for the QBO characteristics to be very close to observations, and for the forcing mechanisms to be unrealistic (ie.: lack of contribution from resolved waves, etc). Hence the addition of metrics addressing the momentum driving of the QBO would be a very important metric to add.
This would be beyond the scope of this paper. See comments to similar points above.

Minor Comments:
1) Page 2, Line 23: There is an inconsistency between 'four CMIP5 models, and 5 CCMVAL models' here and Table 1. In Table 1 only 3 models are listed as part of CMIP5: MIROC-ESM-CHEM, MPI-ESM-MR and HadGEM2-CC ; I believe the CMCCCMS should be included in the list of CMIP5 models in the caption of Table 1. Done.
2) Page 3, Line 3: 'the period of the oscillation. . .' - this should say 'the mean period of the oscillation' - it is well know that the period varies quite a bit as noted further in that paragraph Done.
3) Figure 2 caption: What is hmax ? Added ", where the equatorial QBO Fourier amplitude peaks," to the caption.
4) Page 5, Line 5: 'eleven models' – aren't there only 10 in Table 4? True. Thanks for spotting.

Thank you for your feedback.

---

## Author Comment (AC2) · 12 Mar 2017

[revised manuscript text omitted]

\* The error of these parameters is determined by the grid spacing (refer to Tables 1 and 2).

timeseries.

amplitudes and descent rates are standard errors based on averaging over QBO cycles. The error in the Fourier amplitude, min/max period is based on surrogate

|                  | Height of | Fourier     |              |               | Lowest  |  |
|------------------|-----------|-------------|--------------|---------------|---------|--|
|                  | maximum   | amplitude   | Latitudinal  | Vertical      | Level   |  |
| Model/Reanalysis | (hPa) *   | (K)         | extent (°) * | extent (km) * | (hPa) * |  |
| HadGEM1          | 15        | $0.7\pm0.1$ | 12.9         | 20.0          | 89      |  |
| HadGEM2-CC       | 6         | $1.0\pm0.1$ | 14.4         | 19.2          | 96      |  |
| MIROC-ESM-CHEM   | 7         | $1.4\pm0.0$ | 13.8         | 16.3          | 69      |  |
| MPI-ESM-MR       | 5         | $1.7\pm0.0$ | 15.2         | 20.8          | 82      |  |
| CMCC-CMS         | 5         | $1.1\pm0.1$ | 16.0         | 22.9          | 85      |  |
| EMAC             | 20        | $1.2\pm0.1$ | 15.7         | 17.7          | 85      |  |
| MRI              | 30        | $0.9\pm0.1$ | 15.2         | 19.6          | 97      |  |
| UMSLIMCAT        | 20        | $1.0\pm0.1$ | 13.2         | 21.3          | 85      |  |
| UMUKCA-METO      | 30        | $0.7\pm0.1$ | 12.8         | 19.6          | 113     |  |
| UMUKCA-UCAM      | 30        | $0.8\pm0.1$ | 13.6         | 18.5          | 115     |  |
| ERA 40           | 30        | 1.3 0.1     | 16.2         | 14.2          | 97      |  |
| ERA-Interim      | 30        | 1.3 0.1     | 16.8         | 14.9          | 89      |  |
| MERRA            | 30        | 1.3 0.1     | 16.8         | 14.8          | 88      |  |
| JRA25            | 30        | 1.1 0.2     | 15.8         | 17.4          | 89      |  |
| JRA55            | 30        | 1.3 0.1     | 16.9         | 13.7          | 88      |  |
| CFSR             | 20        | 1.2 0.1     | 17.4         | 15.2          | 85      |  |
| NCEP1            | 30        | 0.8 0.1     | 15.3         | -             | 85      |  |
| NCEP2            | 20        | 0.8 0.1     | 27.7         | -             | 87      |  |

Table 4. Characteristic QBO metrics calculated from the zonal mean temperature. Values for models and reanalyses are listed; there is no comparable observational dataset.

\* The error of these parameters is determined by the grid spacing (refer to Tables 1 and 2).

| ZM Zonal Wind           | Model mean      | Model mean (ex) | Reanalysis mean |
|-------------------------|-----------------|-----------------|-----------------|
| Height of maximum (hPa) | $12.0\pm3.5$    | $13.3\pm4.1$    | $20.0\pm0.0$    |
| Fourier amplitude (m/s) | $15.1\pm3.2$    | $16.3\pm3.5$    | $14.8\pm0.3$    |
| Latitudinal extent (°)  | $19.2\pm1.1$    | $19.7\pm1.2$    | $21.0\pm0.3$    |
| Vertical extent (km)    | $18.5\pm2.2$    | $18.5\pm1.6$    | $16.8\pm1.5$    |
| Lowest Level (hPa)      | $79.1\pm5.2$    | $79.8\pm3.6$    | $87.6\pm1.7$    |
| Mean Period (months)    | $35.9 \pm 11.2$ | $28.9\pm2.8$    | $28.0\pm0.0$    |
| Min Period (months)     | $28.3\pm9.7$    | $23.8\pm1.9$    | $22.7\pm0.2$    |
| Max Period (months)     | $55.2\pm36.9$   | $39.1\pm9.6$    | $35.0\pm0.5$    |
| Amplitude Easterly      | $-33.3\pm3.7$   | $-33.3\pm4.1$   | $-34.0\pm1.9$   |
| Amplitude Westerly      | $18.9\pm5.0$    | $17.5\pm5.2$    | $15.8\pm0.8$    |
| Descent rate Easterly   | $0.6\pm0.2$     | $0.7\pm0.1$     | $0.7\pm0.0$     |
| Descent rate Westerly   | $0.8\pm0.2$     | $0.9\pm0.1$     | $1.3\pm0.1$     |
| ZM Temperature          |                 |                 |                 |
| Height of maximum (hPa) | $16.8\pm10.7$   | $13.8\pm9.9$    | $30.0\pm0.0$    |
| Fourier amplitude (m/s) | $1.1\pm0.3$     | $1.2\pm0.4$     | $1.3\pm0.0$     |
| Latitudinal extent (°)  | $14.3\pm1.2$    | $14.5\pm1.0$    | $16.9\pm0.0$    |
| Vertical extent (km)    | $19.6\pm1.9$    | $18.9\pm1.6$    | $14.5\pm0.7$    |
| Lowest Level (hPa)      | $91.5\pm14.0$   | $86.3\pm10.2$   | $88.0\pm0.7$    |

**Table 5.** Characteristic QBO metrics in reanalyses and models. The mean and  $\pm$  one standard deviation of the metrics in Tables 3 and 3 are shown. The multi-model mean was calculated from all models (\* excluding CMCC-CMS and both UMUKCA models for obvious shortcomings in QBO modelling (Figure 1)), the reanalysis mean from the most recent datasets, namely ERA-Interim, MERRA and JRA55.

---

## Author Response (AR2)

**Topical Editor Decision: Publish subject to minor revisions (Editor review)** (06 Apr 2017) by Richard Neale

Comments to the Author:

It's clear that the comments from the reviewers indicate that the underlying reasons for the choice of diagnostic and method for the calculation of the diagnostic itself have to be made crystal clear (why do we care about the QBO and why did we choose this diagnostic) if this analysis is to be taken up as a standard by the community. The addition of individual model analyses adds significantly to the value of the paper. I also agree it would be useful to summarize what diagnostics are missing from the analysis that could be done as a next step in order to further diagnose the QBO.

Thank you for reviewing the revised manuscript. To make clear what the diagnostics currently do and do not address, two additions were made:

" to characterise the morphology of the QBO" (p. 1, l. 4)

[revised manuscript text omitted]

Middle row: Mean Fourier spectrum (left) of equatorial zonal mean zonal wind. Contours are drawn at 1, 2, 4, 8 and 16 m/s. The Fourier harmonics around 2 years are averaged to give the latitude-altitude QBO amplitude (middle, same contours). From the $\bar{u}$ timeseries at $h_{max}$ (right), the period of each single QBO cycle is calculated and the easterly/westerly amplitudes are identified.

From the latitude-altitude QBO structure, a cross section at the equator (red) is taken to derive the QBO height profile (upper) and one at 20 hPa (blue) for the latitude profile (lower). From the height profile, the vertical extent, the depth $d_{QBO}$ as well as the maximum Fourier amplitude ($u_{max}$) can be identified. The latitude cross section at $h_{max}$, where the equatorial QBO Fourier amplitude peaks, serves to define the latitudinal extent of the QBO.

[Figure]

**Figure 3.** Equatorial $\bar{u}$ profiles in consecutive months for a descending easterly (left) and westerly (middle) shear zone from the FU Berlin observations (1964-1966 cycle). The heights of phase change in each month are shown in red/blue and are displayed in the right panel.

[Figure]

**Figure 4.** Comparison of QBO amplitudes in $u$ (left) and $T$ (right) from models (solid contours) and reanalyses (dotted contours). The colours show the difference models-reanalyses with blue depicting an underestimation by models and red an overestimation.

[Figure]

**Figure 5.** QBO easterly and westerly descent rates in models and reanalyses. The symbols (diamonds for models, circles for reanalyses and triangle for observations) show the mean and standard deviation within each dataset. The filled symbols contribute to the model/reanalysis mean as shown with the black diamond/circle. The dotted line represents equal descent rates for both shear zones as orientation.

[Figure]

**Figure 6.** Timing of phase change in models (blue, excluding CMCC-CMS and both UMUKCA models), FUB observations (green) and reanalyses (red). There are 407/29/39 west-east changes (distribuation of relative occurence in left panel) and 411/28/39 east-west changes taken into account for models/observations/reanalyses.